# Effect of Different Drying Methods on the Nutritional Value of *Hibiscus sabdariffa* Calyces as Revealed by NMR Metabolomics

**DOI:** 10.3390/molecules26061675

**Published:** 2021-03-17

**Authors:** Sengnolotha Marak, Elena Shumilina, Nutan Kaushik, Eva Falch, Alexander Dikiy

**Affiliations:** 1Amity Institute of Food Technology, Amity University Uttar Pradesh, Noida 201313, India; msengno@gmail.com; 2Amity Food and Agriculture Foundation, Amity University Uttar Pradesh, Noida 201313, India; 3Department of Biotechnology and Food Sciences, Norwegian University of Science and Technology (NTNU), 7491 Trondheim, Norway; eva.falch@ntnu.no (E.F.); alex.dikiy@ntnu.no (A.D.)

**Keywords:** *Hibiscus*, herbal processing, metabolic profiling, antioxidant, metabolites, NMR, Hibiscus acid

## Abstract

Red mature calyces of *Hibiscus sabdariffa* were collected from 16 different locations in Meghalaya, India. Samples were processed using shade drying (SD) and tray drying (TD). NMR spectroscopy was used to assess the metabolic composition of the calyces. In this study, 18 polar metabolites were assigned using 1D and 2D NMR spectra, and 10 of them were quantified. Proximate analysis showed that the TD method is more efficient at reducing moisture and maintaining the ash content of the *Hibiscus* biomass. NMR metabolomics indicates that the metabolite composition significantly differs between SD and TD samples and is more stable in TD plant processing. The differences in post-harvest drying has a greater impact on the metabolite composition of *Hibiscus* than the plant location.

## 1. Introduction

Herbal products and their extracts contain several classes of bioactive compounds and are widely used as drugs, cosmetics, and food additives [1]. *Hibiscus sabdariffa*, also known as Roselle or Sorrel, is an annual, herbaceous, medicinal plant from the Malvaceae family. It is one of the most important fiber crops in India, where its consumption exceeds 600 million tons of dry weight per year [2,3]. The food industry uses *Hibiscus* calyces in the production of beverages such as teas or infusions, juices, jams, jellies, and syrups [4,5,6]. In folk or traditional medicine, the water infusion of the *H. sabdariffa* calyx is considered to have diuretic and choleretic properties, act as a mild laxative, aid in bilious conditions, reduce the risk of heart and nerve diseases, and have anti-hypertensive [7,8] and anti-atherosclerotic activity [9]. The calyces contain a wide spectrum of several groups of bioactive compounds, namely organic acids, anthocyanins, polysaccharides, and flavonoids. This composition explains the versatility of *H. sabdariffa* not only as food, but also in the preparation of herbal medicines [6,10,11,12]. Drying of a plant material can be accomplished by different methods. The shade dry method is a traditional method that is often used for drying plant materials. For the drying of flowers, fruits, and roots, a tray dryer is predominantly used [13]. There is no detailed information in the literature that describes the effect of different processing methods on the final nutritional quality of the calyces’ biomass. In this regard, the study by Tham et al. [14] where four drying methods were compared is very relevant. Among the four drying methods, heat pump drying (HP) achieved the highest drying rate, while solar greenhouse drying (SGD) had the lowest drying rate. This study shows that drying is one of the most important factors affecting the color and protocatechuic acid and catechin content. Today, metabolomics is an important field in plant sciences and natural product chemistry [15]. Nuclear Magnetic Resonance (NMR) is one of the main analytical tools for metabolomics. NMR spectroscopy allows the simultaneous detection and quantification of compounds belonging to different chemical classes: sugars, organic and amino acids, flavonoids, alkaloids, terpenoids, and others. This method was applied to the classification and characterization of different medicinal plants, as well as to the detection of the adulteration of plant materials [16,17,18,19]. The objective of this study is to assess the differences in proximate and metabolic compositions of *H. sabdariffa* calyces using NMR metabolomics from plants collected in different locations in India and dried using the shade drying (SD) and tray drying (TD) techniques. The study of phenolic acids, anthocyanins, and carotenoids, which are also found in the *H. sabdariffa* calyx [20,21,22,23,24], was outside the scope of this study.

## 2. Results and Discussion

### 2.1. Proximate Analysis

The proximate composition of the dry *Hibiscus* calyces (SD and TD) is presented in Table 1 and results are discussed further here. Drying reduced the moisture content of the *Hibiscus* calyces. TD samples had the lowest average moisture content of 7.91%, while SD samples had an average moisture content of 10.47%. The higher moisture content in the SD sample was probably due to the high and controlled temperature of the tray drier. Likewise, TD resulted in a higher ash and lipid content of 6.09% and 5.75%, respectively, while SD had contents of 4.44% and 3.34%, respectively. However, the different drying methods did not have any significant variation (*p* > 0.05) in protein content within samples from the same location. *Hibiscus* calyces processed by SD resulted in a higher crude fiber content (16.67%) than TD processing (15.43%). The lower fiber content in TD samples could be due to the degradation of the sample due to high temperature [25]. The results were in good agreement with [26], who reported that oven drying was more efficient than solar and sun in reducing the moisture and increasing fat and ash content.

### 2.2. Assignment of Resonance Signals of the Metabolites

NMR spectra of thirty-two water extracts from the *H. sabdariffa* calyx were acquired and analyzed in this study. Normalized ^1^H NMR spectra of the same sample dried using the two different techniques are shown in Figure 1. Resonance signals of 18 polar metabolites were identified based on the available metabolomics databases and 1D ^1^H and 2D spectra (see Materials and Methods). In addition, two metabolites were tentatively assigned to 2-oxoglutarate and maltol or its derivatives (metabolites 9* and 18*, Table 2), as shown in Figure 1. The resonance assignment is reported in Table 2.

The most intense signals in the ^1^H NMR spectrum were assigned as follows. The signals at 5.33/87.9 (1H, s), 3.21/44.9 (^1^H, d, J = 17.86), and 2.77/44.9 (^1^H, d, J = 17.86) were assigned to the resonance of hibiscus acid in accordance with the assignment by [27]. Signals at 3.27/56.3 and 3.9/68.8 ppm belong to the three CH_3_ groups and the CH_2_ group of betaine, respectively. The region from 3.3 to 4.3 ppm contains the resonances of glucose and fructose. Succinic and acetic acids (2.41/37 and 1.92/26.6 ppm, respectively) are also considered to be main metabolites. Water extracts of hibiscus samples dried differently generally contain the same set of polar metabolites. A comparison of the absolute integral values of the total spectra from the two drying methods shows that the ^1^H NMR spectra of the TD processed samples are on average 18% larger than the values of the SD processed samples. Therefore, more compounds were extracted from the TD samples.

### 2.3. Principal Component Analysis (PCA)

A principal component analysis (PCA) was performed to determine if the different drying techniques caused significant differences in the metabolite composition of *Hibiscus* calyces. Each resonance signal in all ^1^H NMR spectra was integrated using the protocol described in the Materials and Methods section. The resulting PCA created using a reduced buckets number (255) is shown in Figure 2. The main components PC1 and PC2 describe 88% of the total changes. Distinct sample grouping based on the drying method can be seen in the score plot. TD samples are horizontally distributed in the upper left segment of the scores plot.

Most of the SD samples are located at the bottom right of the scores plot. However, SD samples 15, 14, 16, and 1 are located closer to the TD group.

The vertical distribution in the PCA plot is due to nine variables (buckets) located at the bottom of the correlation plot. All these variables, besides the bucket at 6.52 ppm, have a high correlation coefficient (r) with each other (Table 3). The tentative resonance assignment was done using 2D NMR spectra. The resonances at 6.44, 6.46, 7.65, 7.66, 7.68, and 7.69 ppm were tentatively assigned to maltol or its derivatives. The characterization of different maltol derivatives in water by NMR spectroscopy has been carried out recently [28]. Two vicinal hydrogens in the maltol moiety were assigned as H5 (6.43/117.7 ppm; d; J (5.6 Hz)) and H6 (7.92/158.5 ppm; d; J (5.6 Hz)). Analogously, resonances at 6.46/117.5 ppm (d; J (5.6 Hz)) and 7.66/149.2 ppm (d; J (5.6 Hz)) in the NMR spectra of *Hibiscus* water extracts were tentatively assigned to the maltol derivatives using ^1^H-^1^H TOCSY, ^1^H-^13^C HSQC Heteronuclear Single Quantum Coherence Spectroscopy), and JRES (J-resolved Spectroscopy) NMR spectra. The singlet at 6.52/138 ppm (and the relative bucket) was assigned to fumaric acid.

### 2.4. Quantification of Metabolites

Metabolite quantification was performed using a quantitative NMR approach. The main metabolites found in all extracts were hibiscus acid, glucose, fructose, betaine, acetate, and succinate (Table 4). In addition, the concentrations of γ-aminobutyric and fumaric acids were detected in all extracts. The comparison of the metabolic content of SD and TD samples shows that the TD powders generally have more metabolites, and the absolute average value of the integrated TD spectra is 17.8% larger than SD. However, this is only a general trend and not all of the metabolites were extracted in similar ways from the powders from the different locations. In the following paragraphs, we will discuss some of the metabolites in more detail.

#### 2.4.1. Hibiscus Acid

Hibiscus acid has already been isolated and crystallized from *H. sabdariffa* [27]. The vascular effects of pure hibiscus acid on the rat aorta was studied in vitro [29]. It was discussed that the observed vasorelaxant effect of hibiscus acid potentially explains the antihypertensive properties of *H. sabdariffa* [29].

Hibiscus acid is the main polar metabolite found in the *Hibiscus sabdariffa* calyx water extract (Figure 1 number 10, Table 2 and Table 3). The average content of hibiscus acid in SD and TD sample exceeds 10% of the total powder weight and is about 12 and 14 g in 100 g of dry powder, respectively. Therefore, the SD drying method results in an almost 15% loss of this compound. The content of this acid in the SD and TD samples varies by 7.1% and 5.8%, respectively. Such variation might be explained by the different plant locations, time of harvest, varying phase of flower maturity, or non-homogeneous processing. The extraction method proposed in this study allows us to obtain a solution with a hibiscus acid concentration of 0.6% and 0.7% (*w*/*v*) for SD and TD processing, respectively.

#### 2.4.2. Betaine

Betaine is a zwitterionic quaternary ammonium compound that is widely found in animals and plants. It is an important human nutrient [30]. It acts as an osmolyte and a source of methyl groups and thereby helps to maintain liver, heart, and kidney health. It was estimated that dietary intake of betaine ranges from an average of 1–2.5 g/day [30]. Betaine has synergistic properties to enhance the flavor of amino acids and functions as a feed enhancer [31].

Betaine was found to be one of the main polar metabolites in the *Hibiscus* water extracts (Figure 1 number 13, Table 2 and Table 3). SD and TD extracts contain 12 and 16 mg of betaine in 100 mL, respectively. TD processing results in a higher yield of this compound: 0.31 g in 100 g of dry powder for TD as compared to 0.24 g in 100 g of dry powder for SD (higher on average by 23%). It should be noted that the distribution of betaine in the samples has two different trends: while samples 1–12 have a significantly higher amount of betaine in the TD powder, the amount of betaine in SD samples from locations 13–16 is mostly the same as in the TD samples.

#### 2.4.3. Sugars

Both fructose and glucose were found in the analyzed extracts (Figure 1 numbers 12 and 16, Table 2 and Table 3). SD and TD extracts contained fructose (an average of 86 and 112 mg in 100 mL of extract, respectively) and glucose (an average of 88 and 123 mg in 100 mL of extract, respectively). Therefore, TD drying better preserves these two carbohydrates. As in the case of betaine, SD samples from locations 13–16 show a different trend and the content of glucose in these samples is significantly higher than in other SD samples.

Different studies report the median glucose detection limit equal to 310–973 mg in 100 mL of water, and recognition thresholds equal to 634–1717 mg in 100 mL of water [32,33]. The amount of both sugars found in *Hibiscus* extracts in this study is below the detection limit, and therefore the extracts will not be perceived by the consumers as sweet.

#### 2.4.4. γ-Aminobutyric Acid (GABA)

γ-Aminobutyric acid (GABA) is a ubiquitous non-protein amino acid. The content of GABA in white tea is 50.5 mg, 6.6 mg in cereal bran flakes, 280 mg in chocolate, and 94.8 mg in barley bran per 100 g of dry product [34]. Okada et al. [35] described the tranquilizing effect of a daily intake of rice germ enriched with GABA (up to 26.4 mg GABA per day) and its positive impact on sleeplessness, depression, and autonomic disorder observed during the menopausal or presenile period.

Several studies reported that the daily intake of 10–80 mg of GABA can reduce high blood pressure [36,37].

The distribution of γ-aminobutyric acid (GABA) within the analyzed samples does not follow the same trend as the other compounds (Table 4). Generally, SD samples contain more of this compound, with the exception of the samples from locations 13–16. The average content of GABA in SD and TD samples is 80 and 60 mg in 100 g of dry powder, respectively.

#### 2.4.5. Succinic, Acetic, and Fumaric Acids

Succinic and acetic acids were found in all the samples (Table 2 and Table 3). The average content of succinic acid in SD and TD samples is 135 and 163 mg in 100 g of *Hibiscus* powder, respectively. SD and TD dry powder contain 157 and 164 mg of acetic acid in 100 g of *Hibiscus* powder, respectively. Fumarate is not a main metabolite found in the studied extracts. However, the PCA shows that the concentration of this metabolite significantly differs between SD and TD samples. The performed quantification shows that while some SD samples contain more fumarate than TD samples, other samples contain a similar amount of this compound. TD drying yields a more stable content of fumarate (24 ± 6.5%), while variability within the SD samples is larger (24.6–32%).

#### 2.4.6. Methanol

Methanol was found in all analyzed samples; 1.4 and 1.0 mg in 100 mL of SD and TD extracts, respectively (Table 2 and Table 3). This compound naturally occurs in fruits and vegetables as a pectin component and can be released during juice, puree, or wine preparation. Freshly squeezed fruit and vegetable juices contain 1.2 to 20.0 mg of methanol in 100 mL, and the analyzed *Hibiscus* extracts contain a relatively low amount of this compound in comparison [38,39].

### 2.5. Influence of Drying Methods on the Metabolic Composition of H. sabdariffa Calyces

The quantification of the main metabolites allowed us to estimate the average yield of extracted metabolites, which was equal to 16.1 and 19.5 g in 100 g of SD and TD dry powders, respectively.

It can be concluded that the TD method produced samples with a more stable metabolic composition. Indeed, the average standard deviation of the SD metabolites was higher than for TD processing (18.1% and 8.4%, respectively) (Table 4). Therefore, the TD method is preferable for the production of a high-quality calyces biomass with a relatively stable metabolic composition for pharmaceutical uses. The radar presentation of the calyces’ compositions is shown in Figure 3. It can be seen that the sugar content of SD samples is reduced compared with TD samples. At the same time, the amount of GABA, fumarate, and methanol is higher in SD samples. The analysis of the correlation coefficient, r, between the glucose and fructose bucket integrals shows that these integrals are negatively correlated with maltol (r = −0.54). Therefore, the decrease in the amount of sugars in SD samples might be explained by the formation of some maltose derivatives during SD processing. In addition, SD processing causes an increase in fumarate and GABA. These compounds are known to be the final products of fermentation processes [40]. As discussed previously, the SD process is not so effective at reducing the water content. Wet, warm calyces might be a favorable environment for some microorganisms and fermentation processes. Therefore, TD processing can produce samples with a more stable composition.

### 2.6. Influence of Sample Location on the Metabolic Composition of H. sabdariffa Calyces

Within the samples dried with the same method, the metabolite composition varies between samples from different locations. As discussed above, the composition of the TD samples has a lower standard deviation than the SD-treated samples. Therefore, processing method has a higher impact on the metabolic composition of dried *Hibiscus* calyces than plant location.

SD samples from locations 13–16 should be discussed separately. As shown in Table 4 and the radar presentation of the calyces’ compositions (Figure 3, gray dashed lines), this group of samples has a metabolic composition that is rather different from the other SD samples—a higher sugar and betaine content and a lower amount of GABA, fumarate, and methanol. As discussed in Section 3.5, the decrease in sugar content and the increase in GABA and fumarate during SD treatment might be explained by the possible fermentation of the calyces biomass during drying. Therefore, SD samples can be divided into two groups: samples from locations 1–12 (possible fermentation during drying) and samples from locations 13–16. This division reduces the average standard deviation of the metabolite concentrations for the SD 1–12 group to 10.5%, and for SD 13−16 samples to 7.7% (compared to the 18.1% average standard deviation for all SD samples).

The most probable cause of such a grouping is deviations during SD processing, since the metabolic composition of TD samples from locations 1–12 and locations 13–16 are rather similar. However, the location of *Hibiscus* growth affects the metabolic composition of the samples to a lesser extent than the processing method.

## 3. Materials and Methods

### 3.1. Reagents

Deuterium oxide (D_2_O, 99.9%) from obtained from Cambridge Isotope Laboratories Inc. (Andover, MA, USA), 3-(trimethylsilyl)-propionic-2,2,3,3-d4 acid sodium salt (TSP, 98 atom % D) was obtained from Armar Chemicals (Dottingen, Switzerland), and sodium hydroxide was obtained from Sigma-Aldrich. Ethanol, methanol, chloroform, H_2_SO_4_, and NaOH were of analytical grade and procured from Rankem (New Delhi, India).

### 3.2. Plant Materials

Fresh red calyces of *H. sabdariffa* were manually collected from Meghalaya, India. The same variety of *H. sabdariffa* was planted in the University field located in Amity University, Uttar Pradesh, India. At collection, 16 fresh samples were numbered consecutively from 1 to 16 and the detailed locations from where the samples were collected are provided in Table 5. The distance between location 1 and locations 2–16 was 1350 km (Figure 4). The samples were collected in zip lock bags and further dried using different drying techniques.

### 3.3. Drying Process

Shade and tray drying methods were employed in this study. About 50 g of fresh calyx from each location was used for each of the drying processes.

#### 3.3.1. Shade Drying (SD)

This drying process was performed in the shade, away from sun exposure, for 4 days until a constant weight was obtained. The temperature was recorded as 26–28 °C. To ensure uniformity and even drying, the calyces were flipped once a day. The weight of the samples was measured until a constant weight was reached.

#### 3.3.2. Tray Drying (TD)

The samples were kept in a tray and placed in a tray drier maintained at 60 °C for 6 h. The weight of the samples was measured until a constant weight was reached. The samples were manually observed to see if drying was complete (crushable by hand). The resulting 32 dry samples (SD-1 to SD-16 and TD-1 to TD-16) were crushed using a mixer–grinder. The powdered samples were packed in an airtight container and stored in a refrigerator.

### 3.4. Proximate Analysis

Proximate composition was assessed according to standard AOAC (Association of Official Analytical Chemists) methods [41].

#### 3.4.1. Moisture

About 3 g of ground sample was accurately weighed into a stainless-steel dish and transferred into an air oven that was previously heated to a temperature of 105 °C. The sample was dried for 3 h to obtain a constant weight. The dish was removed from the oven and cooled in a desiccator. The final weight of the sample was taken, and percent moisture was calculated using the following Formula (1):(1)$$Moisture\;\left(\%\right)=\frac{W_{S}\;-\;W_{D}}{\;W_{S}}\times 100\%$$
where *W_S_* is the weight of the sample before drying (g) and *W_D_* is the weight of the sample after drying (g).

#### 3.4.2. Ash Content

Ash was determined by incinerating the sample in a muffle furnace. Approximately 2–5 g of sample was weighed in pre-weighed crucibles. The samples were incinerated at 550 °C in a muffle furnace until a grey/white ash was obtained. The residues were cooled in a desiccator and the weights were noted. The percentage ash content was calculated using the following Formula (2):(2)$$Crude\;Ash\;\left(\%\right)=\frac{W_{C+A}\;-\;W_{C}}{\;W_{S}}\times 100\%$$
where *W_C+A_* is the weight of the crucible and ash (g), *W_C_* is the weight of the crucible (g), and *W_S_* is the weight of the sample before incineration (g).

#### 3.4.3. Crude Fiber

About 2 g of sample was boiled with a 200 mL H_2_SO_4_ (1.25%) for 30 min. The mixture was filtered using filter paper placed over a Buchner funnel. The residue was thoroughly washed with boiling water until acid free. The process was repeated using 200 mL of NaOH (1.25%), after which it was washed with 10 mL of 95% ethanol. The residue obtained was scooped into a clean, dry crucible and dried in an oven at 550 °C for 1 h. The dried sample was cooled and weighed. The dried residue was ignited in an electric muffle furnace (approximately 20 min), cooled, re-weighed, and the difference in weight was calculated. The percentage crude fiber was calculated using following Formula (3):(3)$$Crude\;Fibre\;\left(\%\right)=\frac{W_{DC+S}\;-\;W_{IC+A}}{\;W_{S}}\times 100\%$$
where *W_DC+S_* is the weight of the dry crucible and sample (g), *W_IC+A_* is the weight of the incinerated crucible and ash (g), and *W_S_* is the weight of the sample (g).

#### 3.4.4. Crude Protein

Crude protein was determined using a CN (Carbon Nitrogen) analyzer (Carlo Erba NA-1500 CN elemental analyzer). About 2–4 mg of dried sample was weighed out in a tin capsule with an accuracy of 4 decimals using an analytical micro-balance and forceps. The tin capsules with weighed samples were shaped into little round balls with the help of forceps and placed in a heat cabinet at 60 °C for a minimum of 24 h. After drying, the sample was placed in the desiccator. Crude protein (CP) was calculated by multiplying the total nitrogen content (%) by a factor of 6.25 which is used for the plant material.

#### 3.4.5. Total Lipid Content

Lipid content was determined following the method of Bligh and Dyer [42]. About 50–60 mg of dried powdered sample was homogenized for 1 min with 0.8 mL distilled water, 2 mL methanol, and 1.0 mL chloroform. Then, 1 mL of chloroform was added to the mixture and after homogenizing for 20 s, 1 mL of distilled water was added, and homogenizing continued for another 20 sec. The mixture was then centrifuged for 10 min at 5000 rpm. Two distinct layers were observed, with lipid-containing chloroform at the lower phase and the upper phase containing polar molecules. The lipid-containing chloroform at the lower phase was transferred to a glass centrifuge tube with a pointed bottom pipette; 0.5 mL of lipid extract was evaporated to dryness in a tare glass tube and the weight of the lipid residue was determined. Evaporation, facilitated by a stream of N_2_, was carried out in an evaporation unit at 600 °C and the residue was dried in a vacuum desiccator. The dry weight of the residue was determined to obtain the lipid content of the sample.

### 3.5. Extraction of Metabolites for NMR Analysis

Samples (0.2 g) were covered with 4 mL of boiling water and left for 5 min at room temperature for the extraction. The extract was then vortex and centrifuged for 10 min, and the supernatant was collected. The pH of the supernatant was made up to pH 7.0 using 9 M NaOH.

### 3.6. NMR Sample Preparation

Extract (540 µL) was mixed with 60 µL of 1 mM TSP in 20 mM sodium phosphate buffer, pH 7, in D_2_O. The samples were centrifuged for 5 min at 20 °C and 20,000× *g*. The supernatant (530 µL) was transferred into a standard 5 mm NMR tube.

### 3.7. NMR Data Acquisition and Processing

NMR spectra of all extracts were acquired at 300 K on a Bruker Avance 600 MHz spectrometer equipped with a 5-mm z-gradient TXI (H/C/N) cryoprobe. The experimental settings are shown in Table 6. The spectra were processed using the TopSpin 3.5 software (Bruker, Germany). The signal of the external TSP standard was used for spectral calibration at 0 ppm.

Resonance signals were assigned using our previous data and published reference standards [43,44,45]. Signal assignment was carried out using one-dimensional (1D) ^1^H and two-dimensional (2D) NMR spectra, including homonuclear total correlation spectroscopy (^1^H-^1^H TOCSY), J-resolved spectroscopy (JRES), ^1^H-^13^C heteronuclear single quantum coherence (^1^H-^13^C HSQC) and heteronuclear multiple bond correlation (^1^H-^13^C HMBC) spectra.

### 3.8. Metabolite Quantification

The signal-to-noise ratio (S/N > 10) was used to evaluate if the resonance signals could be integrated. The final concentration used for the quantification of the metabolites was the average value of three integrations of the same resonance signal. The standard deviation of the three integrations of the same resonance (%) was used to estimate the uncertainty of the metabolite quantification.

The total metabolic content of SD and TD powders was estimated by the comparison of the absolute integral’s values in the ^1^H NMR spectra. To achieve this, ^1^H NMR spectra were normalized to the average weight of the sample (0.2 g), the average volume of the water extract (4 mL), and the regions from 10 to 0.08 ppm with the exclusion of the water region (5–4.63 ppm) were integrated using the Amix-Viewer software.

To assess the variation in the concentration of metabolites within the same drying process, the standard deviations in the percentages from the average value for one treatment among 16 samples was estimated.

### 3.9. Statistical Data Analysis

#### 3.9.1. Principal Component Analysis (PCA)

A PCA of the spectroscopic data was performed to determine the differences in the metabolic profiles between the differently dried calyces as follows. First, all the spectra were normalized to the average weight of the extracted powder (0.2 g) and the volume of water used for the extractions (4 mL). The spectral region 0.06–10 ppm was manually binned into non-regular buckets in Amix-Viewer for a total of 331 buckets (Bruker, version 4.0). The Table containing the integral values of the defined buckets for all samples was further transported to SPSS Statistics (IBM, version 25) for an ANOVA analysis. The ANOVA analysis defined which of the buckets (variables) significantly contributed to the differences between the two groups of samples (SD and TD) with the setting of *p* < 0.05 as the level of statistical significance. The samples were divided into two groups: SD and TD. The buckets for which the *p*-values exceeded 0.05 were removed from the following PCA analysis. The reduced table containing 255 significant buckets was further imported to Unscrambler X (CAMO Software AS, version 10.5) where the bivariate correlation analysis was performed.

#### 3.9.2. Bivariate Correlation Analysis

The bivariate correlation analysis was carried out using Unscrambler software to verify if the variables (buckets) have a linear relationship. The Table containing the integral values of the 255 buckets of all ^1^H NMR spectra was used as an input. A correlation coefficient, r, was used to define the strength of the linkage between two variables (buckets).

## 4. Conclusions

The composition of herbal material might vary due to different plant locations, plant maturity, irresponsible or inaccurate collection of material, or differences in post-harvest processing. In this study, we analyzed how two different drying methods and plant location influence the proximate and metabolic composition of dry *Hibiscus* calyces using an NMR metabolomics approach. Thirty-two ^1^H NMR spectra of water extracts of dry *Hibiscus* calyces from 16 different locations were acquired and processed for metabolite quantification and statistical analysis of the spectroscopic data. Our study allowed to conclude that generally, the tray drying method is more efficient than shade drying in reducing moisture, maintaining the metabolite and ash content of the *Hibiscus* biomass. The statistical analysis (PCA) showed that the metabolite composition significantly differs between SD and TD samples and is more stable in tray-dried (TD) calyces. The differences in post-harvest drying have more impact on the metabolic composition of *Hibiscus* than the plant location. These data are of relevance for the industry as compositional stability and reproducibility are required for the further nutraceutical use of *Hibiscus*. Our results point to the need for the standardization of post-harvest processing of herbal material that is to be used as drugs or functional food.

The *Hibiscus* NMR spectroscopy shows the concentration of polar metabolites in a complex mixture. NMR metabolomics was found to be an important and relevant tool in the assessment of the efficiency of herbal drying methods and the metabolic composition of the resulting *Hibiscus* biomass.

## Figures and Tables

**Figure 1 molecules-26-01675-f001:**
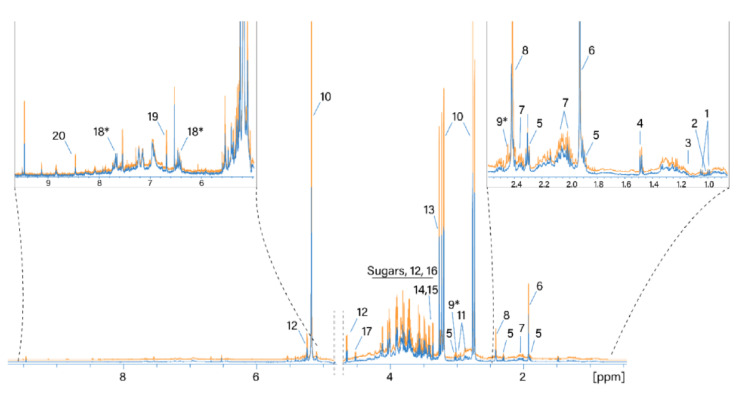
Normalized ^1^H NMR spectra of water extracts of SD- and TD-processed *H. sabdariffa* calyces. Blue: SD samples; orange: TD samples. 1: Valine; 2: isoleucine; 3: ethanol; 4: alanine; 5: GABA; 6: acetate; 7: proline; 8: succinate; 9: 2-oxoglutarate*; 10: hibiscus acid; 11: asparagine; 12: glucose; 13: betaine; 14: scyllo-inositol; 15: methanol; 16: fructose; 17: arabinose; 18: maltol*; 19: fumaric acid. *: Tentative assignment.

**Figure 2 molecules-26-01675-f002:**
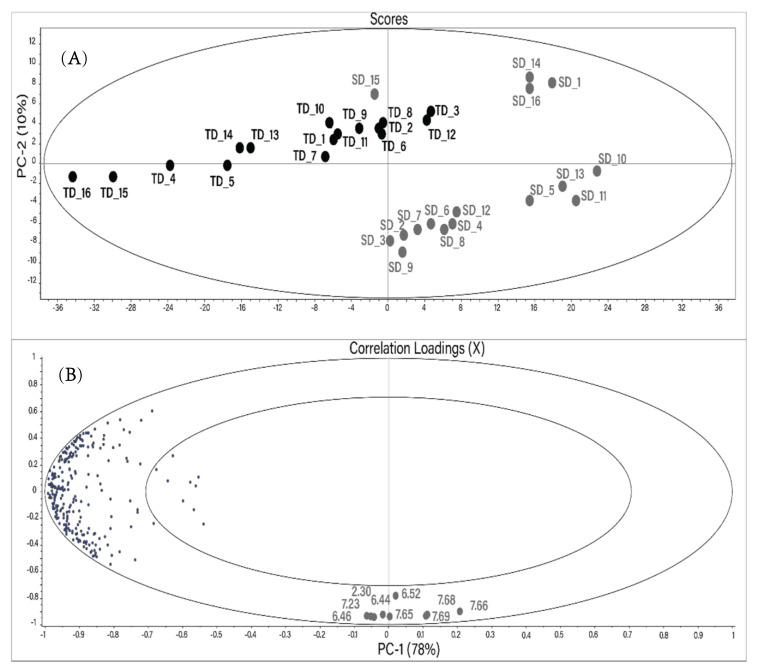
Score plot (**A**) and correlation loadings (**B**) of the mean-centered principal component analysis (PCA) performed using the reduced bucket table of ^1^H NMR spectra of the water extracts of *Hibiscus* calyces. Gray: SD; black: TD. The numbers in the correlation plot indicate the chemical shift of the buckets (ppm).

**Figure 3 molecules-26-01675-f003:**
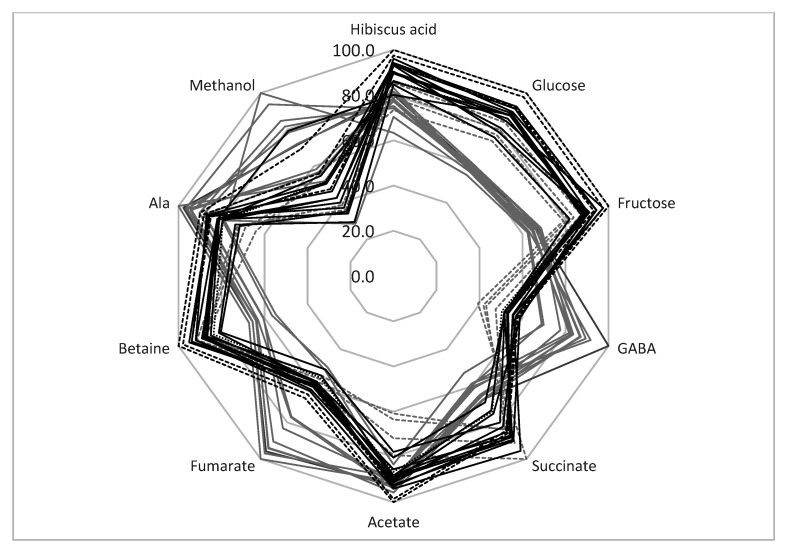
Comparison of *H. sabdariffa* calyx samples (%). Grey: SD; black: TD. Samples 13–16 are shown in dashed lines.

**Figure 4 molecules-26-01675-f004:**
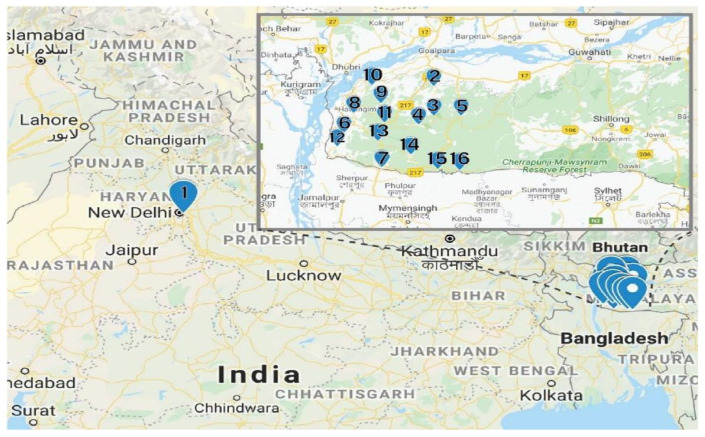
Sample locations. The Garo Hills Region is shown in the insert (Google map).

**Table 1 molecules-26-01675-t001:** Proximate composition of *H. sabdariffa* calyx. SD: shade dried; T.D: tray dried.

Location	Moisture Content (%)		Ash Content (%)		Crude Fiber (%)		Crude Protein (%)		Lipid Content (%)	
	SD	TD	SD	TD	SD	TD	SD	TD	SD	TD
1	9.10 ± 1.50	6.11 ± 0.19	4.88 ± 0.38	5.10 ± 0.38	6.29 ± 0.28	4.82 ± 0.31	8.33 ± 0.45	8.18 ± 0.60	2.32 ± 0.36	3.99 ± 0.55
2	5.93 ± 1.55	2.99 ± 0.52	5.16 ± 0.53	6.29 ± 0.05	11.87 ± 0.49	10.51 ± 0.10	8.21 ± 0.36	7.65 ± 0.17	3.33 ± 0.44	6.77 ± 1.08
3	8.34 ± 0.90	4.65 ± 0.66	4.50 ± 0.05	5.44 ± 0.87	16.38 ± 0.04	15.07 ± 0.04	7.73 ± 0.07	8.70 ± 0.58	4.79 ± 0.76	7.84 ± 1.09
4	7.57 ± 1.44	4.72 ± 0.77	4.60 ± 0.04	6.33 ± 0.40	19.53 ± 0.01	19.02 ± 0.20	7.88 ± 0.20	6.96 ± 0.48	4.57 ± 0.88	7.54 ± 0.01
5	14.0 ± 2.01	8.66 ± 0.57	4.36 ± 0.38	6.43 ± 0.40	17.65 ± 0.03	16.25 ± 0.03	8.24 ± 0.04	8.41 ± 0.71	2.19 ± 0.27	3.78 ± 0.21
6	9.10 ± 1.50	7.20 ± 1.09	4.33 ± 0.57	5.66 ± 0.57	18.65 ± 0.03	18.22 ± 0.01	8.12 ± 0.26	8.05 ± 0.65	3.20 ± 0.24	6.02 ± 0.76
7	12.74 ± 0.18	10.66 ± 0.11	4.43 ± 0.16	6.29 ± 0.25	19.04 ± 0.01	17.85 ± 0.03	8.49 ± 0.3	9.33 ± 0.79	3.57 ± 0.14	6.83 ± 0.31
8	15.13 ± 0.11	15.03 ± 0.16	4.77 ± 0.68	6.51 ± 0.48	16.56 ± 0.02	16.26 ± 0.05	8.00 ± 0.12	8.38 ± 0.61	2.94 ± 0.90	7.65 ± 0.04
9	7.97 ± 1.25	5.88 ± 1.18	4.07 ± 0.13	6.07 ± 0.07	19.18 ± 0.10	17.92 ± 0.04	8.60 ± 0.22	7.37 ± 0.35	3.96 ± 0.39	34.00 ± 0.23
10	12.29 ± 0.62	10.77 ± 0.73	4.36 ± 0.55	6.95 ± 0.07	15.76 ± 0.32	13.81 ± 0.12	8.00 ± 0.62	8.08 ± 0.37	3.51 ± 0.13	5.92 ± 0.16
11	11.24 ± 0.26	9.16 ± 0.88	4.13 ± 0.06	6.01 ± 0.02	18.64 ± 0.10	18.30 ± 0.20	8.05 ± 0.83	7.12 ± 0.08	2.20 ± 0.31	3.66 ± 0.22
12	10.17 ± 0.14	7.00 ± 0.11	4.45 ± 0.40	6.02 ± 0.13	16.44 ± 0.05	14.89 ± 0.10	7.77 ± 0.50	7.62 ± 0.63	4.95 ± 0.41	7.14 ± 0.41
13	15.23 ± 0.20	11.18 ± 0.23	4.24 ± 0.04	6.02 ± 0.04	17.02 ± 0.05	15.92 ± 0.05	8.49 ± 0.17	9.09 ± 0.23	3.52 ± 0.08	5.94 ± 0.28
14	12.50 ± 0.48	10.45 ± 0.30	4.16 ± 0.15	6.65 ± 0.09	15.50 ± 0.07	13.32 ± 0.08	7.04 ± 0.08	5.66 ± 0.66	3.63 ± 0.13	5.38 ± 0.21
15	8.44 ± 0.73	6.03 ± 0.83	4.33 ± 0.57	5.99 ± 0.01	18.73 ± 0.13	17.76 ± 0.19	6.73 ± 0.10	5.52 ± 0.10	2.11 ± 0.26	4.04 ± 0.54
16	7.77 ± 0.39	6.01 ± 0.29	4.33 ± 0.57	5.66 ± 0.57	19.44 ± 0.27	16.90 ± 0.03	7.77 ± 0.56	7.93 ± 0.04	2.69 ± 0.77	5.55 ± 0.38
Average	10.47 ± 0.83	7.91 ± 0.54	4.44 ± 0.33	6.09 ± 0.28	16.67 ± 0.13	15.43 ± 0.10	7.96 ± 0.31	7.75 ± 0.44	. 3.34 ± 0.41	5.75 ± 0.41

**Table 2 molecules-26-01675-t002:** Resonance assignment of 1D ^1^H and 2D NMR spectra of the *H. sabdariffa* calyxes’ water extracts.

N	δ. ppm	Mult.	*J*	HSQC	Assign.	Compound
1	0.99	d	7.00	19.9	CH3	Valine
2	1.02	d	6.90	17.1	CH3	Isoleucine
1	1.04	d	7.00	20.2	CH3	Valine
3	1.19	t	7.10	19.5	CH3	Ethanol
4	1.48	d	7.13	19.1	CH3	Alanine
5	1.89	m		26.5	CH2	GABA ^#^
6	1.93	s		26	CH3	Acetate
7	1.99	m		26.5	CH2	Proline
7	2.06	m		31.6	CH	Proline
5	2.3	t	7.38	37.3	CH2	GABA
7	2.34	m		31.7	CH	Proline
8	2.41	s		37	2(CH2)	Succinate
9	2.46	t	6.87		CH2	2-oxoglutarate *
10	2.77	d	17.86	44.9	CH	Hibiscus acid
11	2.86	m	8.30	37.6	CH	Asparagine
11	2.96	m	8.30	37.6	CH	Asparagine
9	3.01	t	6.87		CH2	2-oxoglutarate *
5	3.02	t	7.50	42.3	CH2	GABA
10	3.21	d	17.86	44.9	CH	Hibiscus acid
12	3.25	m		77.2	CH	Glucose
13	3.27	s		56.2	3(CH3)	Betaine
14	3.35	s		76.5		Scyllo-inositol
15	3.37	s		51.9	CH3	Methanol
12	3.38–3.86					Sugars
12	3.89	m			CH	Glucose
13	3.9	s		69.2	CH2	Betaine
16	3.99–4.04	m				Fructose
16	4.12	d	3.71	78		Fructose
17	4.52	d	7.78	99.8	CH	Arabinose
12	4.64	d		98.9	CH	Glucose
10	5.17	s		87.9	CH	Hibiscus acid
12	5.24	d		95.2	CH	Glucose
17	5.25	d		overlap		Arabinose
18	6.5–6.37	m		117.3–117.6		Maltol *
19	6.52	s		138.3		Fumaric acid
18	7.75–7.62	m		149		Maltol *
20	8.45	s				Formic acid

N: number on signal in Figure 1; δ. ppm: chemical shift in parts per million (ppm); Mult.: multiplicity; J: coupling in hertz; s: singlet. d: doublet. t: triplet. m: multiplet; HSQC: carbon chemical shift of the correspondent cross-peak in the HSQC spectrum.; assign.: assignment; *: tentative assignment. ^#^ γ-Aminobutyric Acid (GABA).

**Table 3 molecules-26-01675-t003:** Significant buckets (ppm) and their r correlation coefficient.

	**2.3**	**6.44**	**6.46**	**6.52**	**7.23**	**7.65**	**7.66**	**7.68**	**7.69**
**2.30**		0.97	0.98	**0.92**	0.98	0.97	0.94	0.96	0.96
**6.44**	0.97		1.00	**0.92**	0.99	0.98	0.97	0.98	0.98
**6.46**	0.98	1.00		**0.90**	0.99	0.98	0.95	0.97	0.97
**6.52**	**0.92**	**0.92**	**0.90**		**0.90**	**0.90**	**0.89**	**0.90**	**0.89**
**7.23**	0.98	0.99	0.99	**0.90**		0.99	0.96	0.98	0.98
**7.65**	0.97	0.98	0.98	**0.90**	0.99		0.96	0.98	0.98
**7.66**	0.94	0.97	0.95	**0.89**	0.96	0.96		0.99	0.99
**7.68**	0.96	0.98	0.97	**0.90**	0.98	0.98	0.99		0.99
**7.69**	0.96	0.98	0.97	**0.89**	0.98	0.98	0.99	0.99	

**Table 4 molecules-26-01675-t004:** Concentration of the main polar metabolites of *H. sabdariffa* calyx (mg in 100 g of dry powder) in shade (S)and tray (T) dried samples.

Sample No.	Hibiscus acid	Betaine	Glucose	Fructose	GABA	Succinate	Acetate	Fumarate	Alanine	Methanol
S1	12,421 ± 82	225.7 ± 1.5	1639 ± 13	1609 ± 7	111.3 ± 5.3	131.6 ± 5.1	169 ± 2	39.59 ± 1.04	20.17 ± 0.00	30.47 ± 0.09
T1	13,119 ± 65	279.8 ± 0.3	2080 ± 5	1971 ± 10	57.7 ± 0.9	134.7 ± 3.1	147 ± 3	24.03 ± 0.33	15.19 ± 0.08	12.50 ± 0.03
S2	11,479 ± 57	219.2 ± 2.1	1662 ± 7	1652 ± 0	92.0 ± 4.0	131.7 ± 2.0	175 ± 0	37.35 ± 1.70	19.94 ± 0.18	39.79 ± 0.12
T2	12,411 ± 51	298.2 ± 4.1	2418 ± 9 ±	2115 ± 14	61.7 ± 1.7	166.4 ± 2.1	161 ± 1	22.47 ± 0.57	16.84 ± 0.08	33.63 ± 0.23
S3	12,621 ± 87	225.5 ± 0.4	1578 ± 5	1594 ± 2	97.0 ± 3.9	128.3 ± 1.5	162 ± 2	40.04 ± 0.40	19.47 ± 0.12	24.72 ± 0.10
T3	14,452 ± 67	316.6 ± 0.6	2508 ± 1	2179 ± 5	65.8 ± 3.3	146.8 ± 10.8	165 ± 0	25.01 ± 0.90	16.90 ± 0.00	18.10 ± 0.08
S4	11,508 ± 76	217.0 ± 1.0	1645 ± 4	1649 ± 3	86.4 ± 3.2	128.7 ± 5.2	172 ± 1	31.89 ± 0.88	18.62 ± 0.27	35.96 ± 0.06
T4	14,011 ± 38	305.4 ± 0.6	2218 ± 5	2193 ± 5	57.4 ± 5.0	174.7 ± 7.7	164 ± 1	25.61 ± 1.13	16.89 ± 0.16	17.01 ± 0.25
S5	12,070 ± 95	220.3 ± 0.9	1609 ± 3	1623 ± 7	93.1 ± 3.2	126.7 ± 3.3	171 ± 1	38.79 ± 0.40	19.36 ± 0.20	33.30 ± 0.00
T5	13,975 ± 15	304.5 ± 1.0	2438 ± 5	2137 ± 2	58.7 ± 2.9	171.5 ± 2.5	170 ± 0	23.90 ± 0.86	16.75 ± 0.25	14.49 ± 0.12
S6	12,896 ± 88	231.4 ± 0.0	1590 ± 5	1521 ± 8	102.1 ± 0.2	114.7 ± 7.1	166 ± 1	41.29 ± 0.88	20.63 ± 0.33	21.99 ± 0.16
T6	14,537 ± 17	316.9 ± 0.2	2322 ± 5	2346 ± 5	62.1 ± 0.7	186.6 ± 2.6	172 ± 0	25.98 ± 1.19	17.94 ± 0.25	21.63 ± 0.21
S7	11,654 ± 88	208.6 ± 2.2	1604 ± 1	1568 ± 13	90.9 ± 0.8	114.7 ± 2.2	169 ± 1	31.73 ± 0.72	18.22 ± 0.16	34.64 ± 0.19
T7	14,435 ± 64	276.7 ± 3.0	2162 ± 6	2172 ± 4	57.6 ± 3.6	156.3 ± 5.7	142 ± 0	21.11 ± 1.63	14.76 ± 0.25	14.93 ± 0.51
S8	10,910 ± 29	194.6 ± 0.7	1567 ± 0	1513 ± 0	77.8 ± 1.2	103.5 ± 5.0	152 ± 0	20.55 ± 0.24	16.42 ± 0.00	12.12 ± 0.87
T8	14,898 ± 53	327.4 ± 0.4	2459 ± 3	2270 ± 9	63.5 ± 1.9	160.7 ± 6.4	167 ± 1	24.41 ± 0.48	17.67 ± 0.25	15.79 ± 0.09
S9	11,976 ± 89	207.9 ± 1.3	1581 ± 3	1551 ± 1	92.2 ± 2.3	119.1 ± 5.4	164 ± 0	34.37 ± 0.32	18.11 ± 0.25	24.03 ± 0.27
T9	13,376 ± 107	292.7 ± 3.8	2358 ± 0	2151 ± 0	59.4 ± 0.8	172.3 ± 12.6	162 ± 0	23.82 ± 1.14	16.72 ± 0.16	21.41 ± 0.25
S10	9811 ± 95	188.6 ± 1.7	1512 ± 4	1500 ± 16	76.7 ± 3.7	116.1 ± 2.2	167 ± 2	21.29 ± 0.16	16.19 ± 0.33	42.41 ± 0.13
T10	14,606 ± 10	323.1 ± 1.2	2539 ± 3	2272 ± 1	65.4 ± 3.3	176.1 ± 19.1	169 ± 0	24.59 ± 0.46	17.76 ± 0.25	19.64 ± 0.09
S11	12,136 ± 5	219.9 ± 1.5	1630 ± 0	1597 ± 2	96.4 ± 2.2	132.1 ± 7.0	170 ± 1	38.89 ± 1.62	19.84 ± 0.08	30.67 ± 0.12
T11	13,327 ± 40	301.1 ± 2.1	2405 ± 5	2129 ± 2	59.0 ± 0.4	170.4 ± 11.9	156 ± 1	24.85 ± 0.24	16.76 ± 0.16	23.23 ± 0.44
S12	12,641 ± 19	220.1 ± 2.2	1664 ± 3	1636 ± 4	99.5 ± 0.5	117.6 ± 1.8	170 ± 1	38.85 ± 0.60	19.78 ± 0.25	23.80 ± 0.05
T12	13,888 ± 61	307.0 ± 0.4	2407 ± 3	2104 ± 3	57.3 ± 2.1	143.4 ± 9.7	161 ± 1	24.08 ± 0.24	16.39 ± 0.00	18.13 ± 0.01
S13	12,555 ± 38	302.6 ± 0.7	2190 ± 0	2018 ± 1	48.0 ± 2.3	163.4 ± 2.5	116 ± 1	22.41 ± 0.65	14.84 ± 0.08	16.41 ± 0.12
T13	13,376 ± 84	288.1 ± 1.8	2408 ± 3	2120 ± 1	56.1 ± 0.8	145.0 ± 8.5	158 ± 0	22.88 ± 0.78	16.38 ± 0.00	21.83 ± 0.03
S14	13,244 ± 58	314.8 ± 0.8	2301 ± 10	2100 ± 21	52.8 ± 2.4	164.0 ± 11.0	111 ± 0	23.85 ± 0.56	14.97 ± 0.25	15.31 ± 0.24
T14	15,119 ± 36	337.3 ± 2.7	2684 ± 4	2346 ± 2	63.9 ± 1.7	165.1 ± 12.9	180 ± 2	26.69 ± 0.31	18.20 ± 0.17	23.77 ± 0.09
S15	12,226 ± 54	303.3 ± 2.7	2121 ± 4	1953 ± 4	46.5 ± 1.2	178.0 ± 1.9	131 ± 0	23.16 ± 0.41	14.23 ± 0.17	21.86 ± 0.00
T15	15,502 ± 48	344 ± 1.3	2742 ± 1	2407 ± 4	64.5 ± 1.4	167.7 ± 10.8	183 ± 0	27.54 ± 0.26	18.57 ± 0.16	29.47 ± 0.32
S16	11,370 ± 72	301.3 ± 1.0	2018 ± 9	1902 ± 2	43.7 ± 1.6	195.2 ± 4.5	145 ± 1	24.69 ± 0.64	13.21 ± 0.00	23.65 ± 0.01
T16	14,544 ± 26	323.4 ± 1.8	2519 ± 18	2242 ± 2	59.1 ± 0.2	171.2 ± 10.4	170 ± 0	25.02 ± 0.18	17.53 ± 0.17	20.15 ± 0.00
St.Dev. SD (%)	7.1	17.7	14.6	11.4	26.8	19.1	13	24.6	13.6	32.7
St.Dev. TD (%)	5.8	6.3	7.1	5.1	5.3	8.7	6.4	6.5	6	27

St.Dev. SD (%) and St.Dev. TD (%): standard deviation within SD and TD samples in 5 replicates.

**Table 5 molecules-26-01675-t005:** Locations of *H. sabdariffa* calyces sample collection.

Sample	District	Topography	Location	Harvesting Time
1	Noida, Uttar Pradesh	28.5439° N, 77.3333° E	Amity University	November
2	North, Meghalaya	25.8987° N, 90.6019° E	Kharkutta	
3			Resubelpara	
4	East, Meghalaya	25.5672° N, 90.5258° E	Songsak	
5			Williamnagar	
6			Rongjeng	
7	West, Meghalaya	25.5679° N, 90.2245° E	Betasing	December
8			Dalu	
9			Selsella	
10			Dadengiri	
11			Tikirkilla	
12			Tura	
13			Zikzak	
14			Chokpot	
15	South, Meghalaya	23.3301° N, 90.5636° E	Baghmara	
16			Rongara	

**Table 6 molecules-26-01675-t006:** Details of the NMR experiment and relative setting.

Experiment	Pulse Program	d1(S)	NS	RG	SW (ppm)	O1 (Hz)
^1^H	*noesygppr1d*	4	128	57	20	2820
JRES	*Jresgpprqf*	2	16/40	57	20	2820
^1^H-^1^H TOCSY	*Mlevgpphprzf*	2	48	57	20	2820/2820
^1^H-^13^C HSQC	*hsqcetgpprsisp2.2*	2	48/128	203	20/165	2820/11,319
^1^H-^13^C HMBC	*Hmbcgplpndprqf*	2	56/128	203	20/260	2820/18,110

## Data Availability

The data presented in this study are available on request from the corresponding author.

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
