# Peer review of "Effect of Different Drying Methods on the Nutritional Value of Hibiscus sabdariffa Calyces as Revealed by NMR Metabolomics"

_molecules, 2021, doi:10.3390/molecules26061675_

Round 1

Reviewer 1 Report

In this study, two different methods of processing 16 different samples of Hibiscus sabdariffa were compared [Shade and Tray Drying]. Moreover, several polar metabolites were assigned and quantified using 1D and 2D NMR spectra.

The authors may address the following issues to strengthen the manuscript:

Line 262-263. The authors explain the Shade Drying process was performed at a temperature of 27°C. It would be necessary to indicate if it was the same temperature during the four days of the process or put the corresponding temperature interval.

Line 265-266. Why was the Tray Drying process done at 60 ° C? Is there any previous work showing that this is the best drying temperature?

Line 268-271. The authors indicated that “The samples were manually observed to see if drying is complete (crushable by hand)”. The authors must explain why the process was not performed to constant weight.

Author Response

Response to Reviewer 1 Comments

Thank you for reviewing the manuscript and providing valuable comments. We have revised the manuscript as suggested and specific responses are provided below:

Point 1: Line 262-263. The authors explain the Shade Drying process was performed at a temperature of 27°C. It would be necessary to indicate if it was the same temperature during the four days of the process or put the corresponding temperature interval.

Response 1: It was the average temperature of 4 days. We have corrected the manuscript by giving temperature interval. Kindly see line no 290 of the revised manuscript

Point 2: Line 265-266. Why was the Tray Drying process done at 60 ° C? Is there any previous work showing that this is the best drying temperature?

Response 2To avoid the degradation of the natural products in the leaf , 60 ° C is the generally accepted temperature. (Roshanak, S., Rahimmalek, M., & Goli, S. A. (2016). Evaluation of seven different drying treatments in respect to total flavonoid, phenolic, vitamin C content, chlorophyll, antioxidant activity and color of green tea (Camellia sinensis or C. assamica) leaves. Journal of food science and technology, 53(1), 721–729. https://doi.org/10.1007/s13197-015-2030-x):

Point 3: Line 268-271. The authors indicated that “The samples were manually observed to see if drying is complete (crushable by hand)”. The authors must explain why the process was not performed to constant weight.

Response 3: The constant weight method was done in the last when it was observed that the leaves are dried. We have added this in the revised manuscript.

Line no 289- 292 0f the revised manuscript.

Reviewer 2 Report

The publication entitled "Effect of different drying methods on the nutritional value of Hibiscus sabdariffa calyces as revealed by 3 NMR metabolomics" by Sengnolotha Maraket. al. describes the effect of two different methods of drying on metabolic composition of Hibiscus sabdariffa calyces.  

The manuscript covers the results of scientific literature on this subject; the original results are well presented and explained. The materials and methods section describes well all the experimental procedures.

There are a larger number of spelling and grammatical errors; there are also a few aspects of content that should be improved.

Please check the keywords. Usually the words from the title are already included in the search syntax. There is no need to be integrated into the keywords. Therefore, Hibiscus sabdariffa and NMR can be replaced by other keywords.

Please check the text and reformulate when is neccesary. For example

Line 16:” NMR  metabolomics allowed to conclude that the metabolite composition significantly differs between SD”

Line 225  . Therefore, processing has a higher impact 224 on the metabolic composition of dried Hibiscus calyces rhan the plants …

Line 217-check

Line 219-check

Line 234 –check

And so on….until to the end of the manuscript

Line 402 The NMR spectroscopy allowed to determine the concentration of polar metabolites in a complex mixture.

Line 403 NMR metabolomics was found an important tool in the assessment of the efficiency of herbals’ drying methods, the metabolic composition of the resulting biomass, assessment of the  causes of the compositional variability and the estimation of components with possible health effect  of the hibiscus water extracts.

Author Response

Response to Reviewer 2 Comments

Thank you for reviewing the manuscript and providing valuable comments. We have revised the manuscript as suggested and specific responses are provided below:

Point 1: There are a larger number of spelling and grammatical errors; there are also a few aspects of content that should be improved

Response 1:  The manuscript has thoroughly been checked and corrected.

Point 2: Please check the keywords. Usually the words from the title are already included in the search syntax. There is no need to be integrated into the keywords. Therefore, Hibiscus sabdariffa and NMR can be replaced by other keywords

Response 2 :  The Keywords have been changed

Point 3: Please check the text and reformulate when is neccesary. For example

Line 16:” NMR  metabolomics allowed to conclude that the metabolite composition significantly differs between SD”--CORRECTED

Line 225  . Therefore, processing has a higher impact 224 on the metabolic composition of dried Hibiscus calyces rhan the plants …- CORRECTED

Line 217-check-- CORRECTED

Line 219-check- CORRECTED

Line 234 –check-V  CORRECTED

And so on….until to the end of the manuscript- CORRECTED

Line 402 The NMR spectroscopy allowed to determine the concentration of polar metabolites in a complex mixture.- CORRECTED

Line 403 NMR metabolomics was found an important tool in the assessment of the efficiency of herbals’ drying methods, the metabolic composition of the resulting biomass, assessment of the  causes of the compositional variability and the estimation of components with possible health effect  of the hibiscus water extracts.

CORRECTED

Reviewer 3 Report

This paper investigates the impact of drying methods on the proximate and metabolic compositions of Hibiscus sabdariffa calyces collected from different locations in India. The work is interesting. However, I have some observations that are listed in the following lines.

  1. There are some typos and grammatical errors. The work would benefit from close editing.
  2. Throughout the manuscript, avoid starting a sentence with a number that is not written out.
  3. Line 50: what is the reason for excluding phenolic acids, anthocyanins, and carotenoids? The studied drying methods could affect these groups as well. Explain.
  4. Line 51: Results and Discussion
  5. Line 266: The duration mentioned in Table 6 for tray drying is 6 hours while the authors mention that samples were flipped once a day. Explain.
  6. Since the temperatures are already mentioned in the text, I suggest removing table 6. The time duration can be added to the text as well.
  7. Subsections 3.5 - 3.8 describe the same experiment and should be merged.
  8. The conclusion is too long. Rewrite in a more concise way.

Author Response

Response to Reviewer 2 Comments

Thank you for reviewing the manuscript and providing valuable comments. We have revised the manuscript as suggested and specific responses are provided below:

Point 1: There are some typos and grammatical errors. The work would benefit from close editing.

Response 1:  The manuscript has been thoroughly revised.

Point 2: Throughout the manuscript, avoid starting a sentence with a number that is not written out.

Response 2 :  We have taken care of that in the revised manuscript.

Point 3: Line 50: what is the reason for excluding phenolic acids, anthocyanins, and carotenoids? The studied drying methods could affect these groups as well. Explain.

Response 3:  This is outside the scope as their studies require application of a complex structural NMR experiments and their interpretation  would make the paper heavier and would change accent of our work

Point 4: Line 51: Results and Discussion

Response 4: Corrected

Point 5: Line 266: The duration mentioned in Table 6 for tray drying is 6 hours while the authors mention that samples were flipped once a day. Explain.

Response 5: It was mistakenly written. It is now corrected in the revised manuscript.

Point 6: Since the temperatures are already mentioned in the text, I suggest removing table 6. The time duration can be added to the text as well.

Response 6: It is now mentioned in the text and table has been removed.

Point 7: Subsections 3.5 - 3.8 describe the same experiment and should be merged.

Response 7: We agree that they belong to same experiment , however, due to sake of clarity we have made these subsections.

Point 8: The conclusion is too long. Rewrite in a more concise way.

Response 8: The conclusion has been shortened.